# Renewable energy for sustainable development in China: Discourse analysis

**Baohong Jiang[1], Muhammad Yousaf Raza[2]***

**1** School of Foreign Languages, Shandong Technology and Business University, Yantai, Shandong, China,
**2** School of Economics, Shandong Technology and Business University, Shandong, China

* yousaf.raza@ymail.com, yousaf.raza@sdtbu.edu.cn

## Abstract

China is the world's largest renewable energy installer with a capacity of 1020 gigawatts in 2021. This study aims to analyze the public discourse around China's green energy and green technology and the paths to sustainable development by comparing public policy. The public discourse analysis approach and Grey Prediction Model are applied to analyze the motives for the distinct inferences being reached over the influences of renewable energy initiatives (REIs). The findings show that the modeling and assumptions are found different in theoretical perspectives, especially in the case of economic and environmental sustainability. The results are close to the other jurisdictions following REIs, including feed-in-tariff, standards and renewable liabilities. Based on statistics during 2012–2021 Five-year plan period, three major renewables are forecasted under base, reference and aggressive scenarios with interesting results. The wind would rise by 109 terawatt hours in an aggressive scenario while solar will rise from 83–99% with a rise of four times in the next decade. Finally, China's current energy policy has been proven to be a series of effective public policies by making the discourse analysis, which can energetically widen the subsidy funds' sources, discover miscellaneous financing techniques, standardized the subsidy process, supervise in applying the renewable energy technologies, and enhance the feed-in-tariff attraction of consumers and private investors.

**Data Availability Statement:** All relevant data are within the manuscript and its Supporting Information files.

**Funding:** This paper is supported by 2020 Annual Shandong Provincial Government Scholarship

## 1. Introduction

Whether or not China's present policies are adequate to accelerate and encourage the involvement of different energy to the country's overall supply is a key question, especially given China's current energy contribution framework and environmental situation. The current decade guides society ideologically and practically in understanding the requirements, issues and China's future. During the past ten years of development, it can be seen that in the context of overall energy production, it indicates a trend of rising on the whole. Overall energy production of China in 2010 was 3.12 billion tons of standard coal with a rise of 9.1% compared with 2009, which was the highest rise of 4.98 billion tons of standard coal until 2020 [1].

As per the annual meeting of the National People's Congress under the 14th Five-Year Plan (2021–2025), the green energy, green economic and social development until 2035, China

Program for Overseas Study in the Education System, Digital Empowerment of New Media Communication Technology and International User Experience Development Commissioned by Enterprises and Public Institutions (Project Number: 010966).

**Competing interests:** Authors declare there is no conflict of interest.

focused on building green energy infrastructure involving climate and energy policies for the coming decade has been set. The environmental merits of the national congress projected to encourage the broad-level of minor-effect renewable energy technologies (RETs), including solar, wind, small hydropower, electricity, and biogas generation. The Feed-In-Tariff (FIT) announced in 2019 for solar projects that set at the center of legislation contended that it presented the possibility to "benefit the installation caps and FIT in solar and wind" and FIT so-called "village-level" poverty alleviation [1]. The aim is to connect the village and regulate simplicity to create successful industrial growth while limiting prices and lessening and substituting polluted electricity sources with RETs. Also, in the 13th Five-Year Plan on renewable energy, China set a target of 15% non-fossil energy in primary energy utilization, 675 megawatts of renewable power capacity by 2020 and non-fossil energy in primary energy use by 2030 [2]. The plan was employed with substantial feed-in-tariffs, which access to capital from government policies. However, this program will create jobs and enhance the economic development in various provinces, reduce unit production costs, rate of return and employment and make households better off [3].

Assessment of the economic influence of the regulation has had a key role in renewables policy in China as well as in the Belt and Road countries [4]. As per the China Energy Portal [5], renewable energy consumption accounted for 40.8%, showing that the share of clean energy consumption has increased nuclear, hydro, wind, and solar by 5, 4.1, 15.1, and 16.6% in 2020. The provincial government is also efficiently working on rising renewable resources and cost reduction to control pollution [6]. Similar discussions are occurring in other jurisdictions following clean energy initiatives, for example, Denmark, Germany, Nigeria, Ontario, and Russia [7–9], which enthused China legislation.

The energy division is about the major source of greenhouse gas (GHG) emissions and it has been estimated that worldwide, energy-related carbon emissions will rise by 16% by 2040 [10]. Taking into consideration the environmental degradation and discharge of dangerous gases into the air coming from the exploitation and consumption of fossil fuel sources, there is a need for China to expand its energy mix towards its energy needs. Currently, China has a strong reliance on fossil fuels, such as oil, coal and gas [11]. There is a need to raise clean energy applications, such as solar, biogas, hydro, and wind energy in rural and urban energy planning.

The study motivation and contribution present: (i) the study objective is to explore the clear concerns over China's perceived incapability to meet its energy demand and to add to the discourse on the ways to obtain sustainable economic growth through a low-carbon emission pathway. (ii) Based on major renewable energy, such as solar, wind and biomass are employed to develop for China. To estimate renewable technologies of '3' factors, we employed the Grey Prediction Model (GPM) from 2012–2021. To predict the RETS, the forecasting method is collected by a transformed GPM linked with '3' scenarios, including business as usual, reference scenario and aggressive scenario. (iii) The empirical statistics on the economic effect of China legislation are very few. Somewhat the evidence-base for the discussion over the economic influences of clean energy and the environment ran about completely from the outcomes of various economic modeling practices. The results are comparable to those observed in other jurisdictions following renewable energy initiatives (REIs), for example, FITs, and standard and renewable energy obligations. Thus, the study aims to address relative public policy and discourse analysis methods to investigate the motives for the various inferences being reached over the effects of REIs. This will help to potential enhancements in efforts to comprehend the economic impacts of these initiatives in the forthcoming. (iv) The study answers the following questions: what are the impacts of China's green energy and economic growth policy using public policy and discourse analysis? What are the supporters of the

legislation at various findings and practices? What are clean energy obligations and portfolio standards in China? What is the role of the market and state in sustaining economic development and environmental sustainability?

The next part of the study is as follows: section 2 presents the literature review. Section 3 provides methods and energy background; section 4 discusses the results and discussion while conclusion and suggestions are provided in section 5.

## 2. Literature review

Present literature dealing with China's energy, environment and economy, which is divided into different parts. The majority of the well-known literature on energy intensity and its influences emphasizes economic development [12]. Few of the researchers have analyzed the idea that economic development can rightly be decupled from energy intensity by enhancing a continuing countrywide transition to renewables [13], and few of them accept that this transition can lead to an incessant decline in energy [14]. Thus, the association between energy and economic development in emerging nations should be further sightseen, as it is debatable. Besides, economic growth, technological change and development, industrial reform, industrial level, urbanization level, and energy prices are normally referred to as energy factors in the literature on many descriptive factors [15–20]. Energy discourse has still to obtain adequate attention as an imperative cause of growth in energy.

Energy efficiency and renewable energy are the key elements of natural resources in China [21]. Most of the researchers analyzed the importance of various energy resources and costs that persuaded its significant role in human capital, energy and industrial development [22, 23]. Political discourse and the role of governments in national and international issues have been broadly studied in current decades, particularly linked to environmental, social and energy issues [8, 24–28]. In addition, Zerva et al. [29] employed the multicriteria decision analysis method to check the climate change stakeholders in Greece and found that citizens' involvement and sustainability can enhance the pollution; Kyriakopoulos et al. [30] analyzed the electricity use and RES plants in Greece and estimated that energy policy is the best way; Arabatzis and Malesios [31] used the RES in the short and long-run and described that public awareness avoids the environmental risk; Tampakis et al. [32] found the RES sustainability in Greece that contributes to the energy security; Zografidou et al. [33] used the financial method to renewable energy production in Greece and found that social, financial and power aspects are major concerns; Raza and Dongsheng [34] used the carbon source and carbon damage for Pakistan and estimated that renewable energy policies for renewable production should be tailored to local situations.

These studies have searched to estimate the governmental discourses affecting the countrywise energy, environmental and related challenges, reflecting upon the differences and similarities in government policies and methods to these issues to give recommendations to support updated discussions about addressing current challenges. An example of China's governmental discourse approach is provided by Liu and Liu [28], which check media discourse on communal insights into biomass energy consumption, and provide those social influences impact energy system change. Another study by Zhou and Qin [35] analyzed natural gas growth based on a five-year plan. They found that China's natural gas penetration may not be as quick as predicted unless the institutional reform of natural gas is first ranked and more indigenous gas suppliers are brought in to strengthen market competition. Thus, the role has concentrated strongly on the media role and one aspect of energy discourse, government tariff, renewable energy, environment, and economic policies have not typically been discussed in China's business models. In addition, the major concentration has been seen on China's climate change

and mitigation of carbon emissions by applying renewables, for example, developing specific regional emissions reduction targets, level of renewable capacity, incentivizing market for its development, feed-in-tariff, trade system, and subsidies on renewable generation are the major concerns. Moreover, few studies analyzed that long-term mitigation pathways for net-zero emissions for the socio-economic benefits [36–38]. They found that national strategic planning can be enhanced. Thus, the work done by Zhou and Qin [35] and Liu and Liu [28] is so far from our work on this issue. This study uses comparative public policy and discourse analysis to investigate the motives for various inferences being achieved regarding the effect of renewable energy, and to recommend productive enhancements to comprehend the economic influence of these initiatives in the future.

## 3. Approaches and energy background

### 3.1. Energy policy and public discourse analysis approach

Along with the development of discourse theory, the rising scholars have been focusing on the analysis of discourse in public policy. Some scholars pointed out that debating the discourse is central to the process of public policy [39]. Discourse analysis is one of the most widely used approaches to understanding the framework and variation of government's public policies, which helps to seek technical routes to solve public problems. Fairclough [40] as one of the pioneers of critical discourse analysis argued discourses and other aspects of social practice were constituted as a dialectical realization relationship, i.e., discourse is constrained and influenced by other aspects of social practice while also influencing and shaping other aspects of it [41]. Thus, discourse is no longer a mirror image of objective reality, but participates in the construction of social reality as one of the social practice elements. In other words, the discourse analysis approach has become an important entry point for researchers to explore sociocultural and governmental interactions and to enrich research perspectives.

There is no doubt that the increasing politicization of energy issues has become an indisputable fact because its discourse carries the significance of a country's position, viewpoint, ideology, etc. [42]. Energy discourse has been extensively studied in national economic policies and international positions for several decades of years. These studies are analyzed in a literature review that seeks to understand the impact of government energy discourse on the global environment and global energy challenges, to compare the differing impacts of different government policies and approaches to these issues, and thus to develop cutting-edge ideas for addressing these challenges. Since China is now the largest country in terms of energy consumption and greenhouse gas emissions, the progress of energy decarburization in China has a significant impact on the global environment [35]. Therefore, the analysis of China's renewable energy and economic progress from the perspective of public discourse analysis has realistic significance for world energy development. In light of these issues and challenges, this paper takes China's energy economic development as the main line, analyzing the interaction between energy policy discourse and energy practice, and the construction of public discourse on energy economy as the entry point to analysis of the impact of China's energy policy on China's energy direction.

### 3.2 Analytical method

This approach in China has inclined to highlight the roles of government agencies, frameworks and public policy discussions and to the suggestions. In respect of the environment, energy resources and policy, the significance of the physical nature of the environment, energy issues and the economic background within which policy choices are done have also been emphasized [43, 44]. Underlying the ideas, values and assumptions in policy perspective are

commonly recognized in the debate of methods to the research of communal policy [45], the method in which they figure and certain policies have usually received less emphasis in public policy literature.

The methods related to discourse investigation place a transformed focus on the significance of understanding the norms, disagreements and judgments in the policy debates [28]. This kind of method is especially helpful in analyzing the discussion on renewables, where economic analysis practices have given much evidence. The economic modeling is applied for theoretical purposes, which helps to understand the efficiency of various kinds of policy interventions on the economy, society and environment. For this, they employed implicit and explicit assumptions on various modeling approaches, using various factors [46]. These approaches bring various inferences, even inside the general modeling. The assumptions taken in both the model framework and efforts employed for definite modeling practices are expected to replicate the opinions of the researchers entailed, especially in uncertain fields [47]. Thus, the estimation of economic impact, including REIs, market roles, country's share and ecological sustainability in public policy are the major concerns.

The discussions based on the economic growth influences of clean energy could be a major motive, but previous studies have not discussed this matter. Few falls into groupings recognized by Dryzek [48] who discussed environmental discourses, including market fundamentalists, economic rationalists and ecological modernists. All of them are inclined to assist REIs in Europe and North America. As per the study by Su and Fan [49], the advancement of RET and industrial up-gradation in Chinese provinces provided significant outcomes. The range of viewpoints existing in the economic debates on energy and trade can be seen in a triangular form in Fig 1. These may also help main policy interferences in the background of the scale and time frames within which socio-economic and socio-technological transitions are required to modernize sustainability. Therefore, these factors ultimately support comparative to the proficient market's working, societal justice and collaboration with the world.

### 3.3. Renewable energy and green economy

As per the National Renewable Energy Laboratory (NREL) classified renewable energy policies of China into three types [50]. First, the policies give general direction and guidance about the enhancement of renewable energy and its influence on the global environment. Second, policies are based on basic principles, directions and renewables policy objectives. The first two policy levels are well-known and applied by the central government of China. Under these policies, local government, Provencal, district, city, and country governments estimate the third-level policies, which provide particular estimations and local goals to enhance jurisdictions and renewable energy. For example, at the country level, the renewable energy policy of China dates back to the early 1990s. In 1992, China Agenda 21 was implemented to encourage sustainable development of natural resources and climate change. During this period, various kinds of related policies were framed, i.e., the illumination and drive the wind program in1994, the 9[th] five-year plan of industrialization, renewable energy plan in 1996, the energy conserving plan in 1997, the 10[th] five-year plan for commercializing the renewable energy in 2001 [50]. However, the first renewable energy law of the PRC was passed in 2005, which recognized regulations on clean energy about resource surveys, planning, development, technical support, industrialization revolution, cost estimation, expenses, economic encouragements, and supervisory measures [51]. After that, these regulations have increased the development of China's renewable energy. In 2006, the National People's Congress permitted the 11[th] five-year plan from 2006–2010, which stated a list of necessary energy, climate goals and environmental issues. The objective was to lessen energy intensity by 20%, water consumption by 30% and

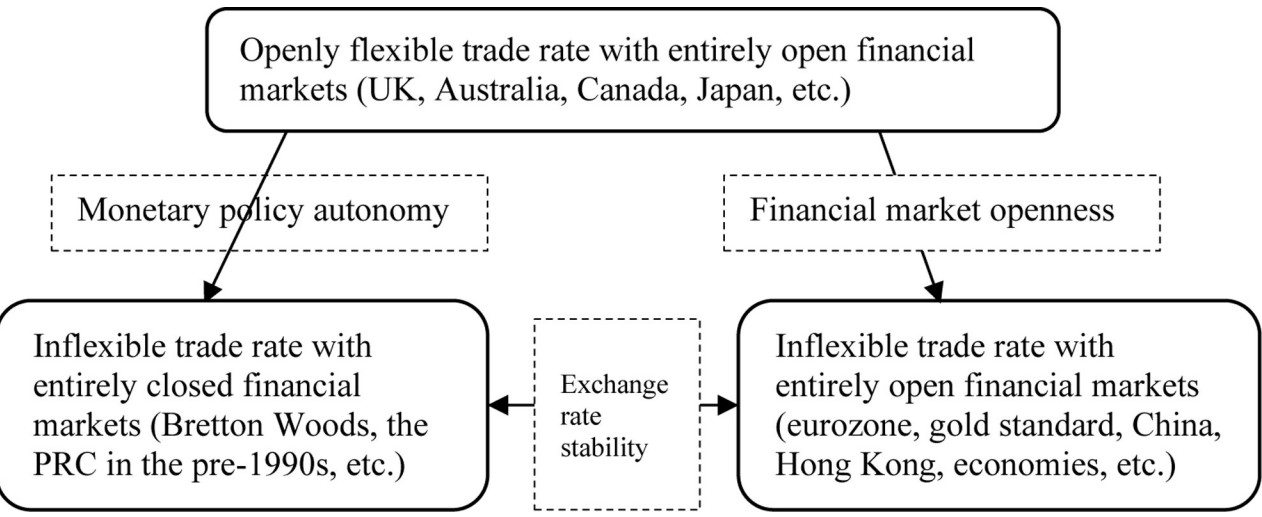

**Fig 1. Economic tolerance for energy-trade market under the People's Republic of China (PRC).**

energy consumption per unit by 20%, $CO_2$ emissions by 10%, and raise the share of renewable energy by 10%. [52]. Currently, green energy and economic development until 2035 bring a lot of development, such as solar, wind, hydro, biogas, electricity, RETs, and FIT [53]. Moreover, the electricity system based on various design features of FIT program is: the FIT rate, domestic content, incentives for the community, technology, regulatory approvals, and environmental benefits, which are provided in Table 1. China's situation for electricity systems was the biggest marketplace for the new volume included in utility-scale solar PV, adding about 35% worldwide in 2021 [54]. China being the biggest new offshore and onshore wind, solar PV, and hydropower market, drives the possible investments by country. Besides, it has a lesser average evaded cost, because of the movement of certain low-cost than-gas and coal-fired plants. Therefore, China should target lessened generation prices in response to USD 31 billion, which is 56% of the global in 2021.

**Table 1. Renewable energy and its impact on China in 2021.**

| Renewable energy type | Generation (GWh) | Capacity (MW) | Cost (USD million) as a whole | FIT price(¥/kWh) 2021 | Patents technology (cumulative) | $CO_2$ emissions avoided (Mt) |
|---|---|---|---|---|---|---|
| Solar PV | 261,639 | 306,403 | 60.35 | 0.49 | 169,460 | 12Mt |
| Concentrated solar power | 19.94 | 570 | | 0.49 | 18,676 | 0.0 Mt |
| Wind offshore | 14,889 | 26,390 | 0.33 | 0.75 | 70,661 | 5.6 Mt |
| Wind onshore | 452,148 | 302,583 | | 0.47 | | |
| Renewable hydropower | 1,321,710 | 354530 | 78.84 | 0.37 | 18,181 | 5.3 Mt |
| Pumped storage | 33,500 | 36,390 | | | 2,462 | |
| Marine | 6.500 | 4.7500 | | | 8267 | 0.0 Mt |
| Geothermal | 143.500 | 25.7500 | 40.29 | | 3,521 | 0.0 Mt |
| Biogas | 4574 | 1711 | 53.90 | 0.75 | 25,118 | 1.8 Mt |
| Renewable municipal waste | 36,522 | 11,022 | | | 19,717 | |
| Solid biofuels | 57,882 | 17,021 | | 0.75 | | |

**Source:** China Energy Yearbook.

As shown in Table 1, the renewable energy supportive structure was recognized; giving for associated ecological support of renewable energy plans, and reducing the supporting costs from the government [44]. China will add 40% of global renewable capacity expansion between 2019–2024, led by enhanced system integration, limited cost and greater competitiveness of both solar PV and onshore wind [55]. The actual FIT rates and the rates as upgraded in 2021 are provided in Table 1. The policy challenge is to guarantee enough investment in electricity networks and in a mix of production technologies that are fit for power system needs. As per the China Energy Portal [5], the long-run renewable energy strategy will continue this goal, but protracted the deadlines for its success to zero emissions.

## 3.4. Development forecast of China's renewable energy using Grey Prediction Model (GPM)

Numerous medium and long-term predicting theories are available, which can be differentiated by the forecasted method, the correctness of outcomes and vice versa. Taking both qualitative and quantitative needs for the forecasting outcomes [56], a grey forecasting method combined with the scenario analysis for the future application of China's renewable energy in the current study. On the basis of renewable energy during the 13[th] and 14[th] Five-year Planning, both the volume to substitute conventional energy and the used area by renewable energy in 2012 and 2021 are predicted using the grey prediction model. Though, wind, solar, and biomass energy are used in China, thus, only renewable energy technologies related to energy were taken in the predicting process. All the related statistics have been collected from China Statistical Yearbook (various issues). For instance, during the 14[th] Five-Year Plan for a modern energy system, the plan involves a target for the non-fossil proportion of electricity to increase by 5.8%-39% which is consistent with the current trend. Under the Grey Prediction Model (GPM), the historical statistics of renewable energy are limited and its relationship and its association with the macroscopic parameters, i.e., GDP and population, etc. are unclear. Thus, employing the GPM, the unidentified factors can be checked as a "black box", which abridges the multifaceted association. However, the forecasted part can be taken as the extension of the present situation, thus, the forecasted process will be comparatively smooth and steady [57]. The GPM is advantageous due to the present growth features of renewable energy concerning a few indefinable factors, i.e., administrative activities that pointedly impact the growing tendency, however, original GPM cannot make instant answers to these factors. In the scenario analysis, we can compensate for the above disadvantage of the GPM to some degree, by constructing various scenarios, those unidentifiable but projecting factors can be considered in to account. According to Meng et al. [58], scenario analysis is a type of instinctive and qualitative forecasted technique that is appropriate for the trend forecast. In the current study, the predicting technique is collected by a transformed GPM joined with scenario analysis.

Using the original GPM model $GPM(1,1)$, $\varphi$ as so called "scene factor" is added, which is the weight coefficient of the joint forecasted method. In the current method, $\varphi$ is the correction factor of the evaluation parameter a, and the region of $\varphi$ is $(0,+\infty)$. The relevant equation of the GPM model $GPM(1,1(\varphi))$ is as follows:

$$y^{(0)}(h) + \varphi a w^{(1)}(h) = b \tag{1}$$

Where $h$ is the grey derivative; $a$ is the evolution parameter; $w$ is white background value; and $b$ is grey action.

## 3.5. Development of scenarios

We set up various scenarios during the prediction process. We used renewable energy factors to check their effects under the 2060 net zero emission plan of China [59]. For this, we used '3' development scenario for renewable energy and carbon emissions in China, such as business as usual (BAU), reference scenario (RS) and aggressive scenario (AS) from 2012–2021. The conditions of each scenario are as follows:

First, in the BAU scenario, there is no consideration of the macro-requirements of country's energy structure alteration and the burden of $CO_2$ emission mitigation from the worldwide society, constructing energy utilization is growing on the basis of current trend, while at the other moment, the expert inclines to be conservative in renewable energy implication. Second, the RS is steady with tradeoff and balances, which widely deliberates different factors, for example, resource potential, ecological limitations, and social cost, etc. It needs energy promotion policies, rational assessment, and self-motivation from the market to meet the predicted demand. Third, in the AS, the traditional energy resources are about exhausted and the substitution of renewable energy resources is at the maximum level. Because of the pressure on national energy security, the government has to maximally raise investment in substitutive energy fields, and more plans concentrate on renewable energy. Hence, the renewable energy investment becomes self-motivated, thus, different RETs, effective operation and management are broadly implemented in different sectors.

# 4. Results and discussion

## 4.1. Results

About $2/3^{rd}$ of all renewable energy careers is based in Asia, with China only adding 42% of the worldwide overall [60]. The global job in renewable energy touched 12.7 million in 2020, a jump of 700,000 new employments in a single year, with China adding 42% of the total [61], which shows the country's strength in installation markets and equipment manufacturing. To protect jobs and related socio-economic benefits, more countries across the globe need to follow policies to encourage their indigenous capabilities. As the number of jobs in cleaner energy continues to increase, it is necessary to guarantee that these posts give decent livelihoods in the form of wages, occupational health, safety, work environment, job security, and other rights.

Practically, being an emerging sector, renewable energy information is limited for the public. Little evidence was established to be held by the Ministry of Energy, World Bank, International Renewable Energy Agency, and Journal articles. Furthermore, there are few subjective and media reports concerning the validation of renewable manufacturing companies [28]. We have seen that overall discussions on jobs, renewable energy, and political debates in the clean energy division in China could be recognized. China's facts on industrial occupation and related FITs, technology, cost, capacity, generation, and carbon mitigation were collected using authenticated engines, which identify consistent results. For example, detailed information based on renewable energy sector employment is accessible in the United States and the United Kingdom [62, 63]. As per the Council of Economic Advisors [64], the United States federal government has created complete valuations of the employment creation influence of renewable energy. Thus, the accessibility of remarkably robust human capital, comprising administrative and management abilities in the state-owned, builds China's economy strong in the face of obstacles [65, 66].

As per data and information on China, the two types of modeling have been taken on energy. First, energy rates will change as per the development of renewable energy generation

size. The relevant studies based on author, methods, analysis design, conclusion, and discursive viewpoints are outlined in Table 2. Second, it measures the economic influences that REIs will have in China, involving its impact on employment creation, economic growth, capital, and environmental issues. According to the second type, economic impacts based on REIs are presented in Table 2. Also, Sebos et al. [67] estimated stakeholder perceptions on climate change influence and application in Greece and found that planning and application measures are crucial in addressing and lowering the influence of climate variation. Similarly, multicriteria analysis was performed by Ioana et al. [67]. Moreover, from a broader perspective, Sebos et al. [68] concentrated on the transformation of school yards in Greece using an interdisciplinary method and found that local communities, urban planners and policymakers play an imperative role in mitigation potential.

In addition, based on given modeling outcomes instead of experiential evidence in structuring the discussion about renewable energy, it becomes more imperative to identify the various assumptions being applied by the modelers in concluding their outcomes. Based on the key question is the matter of forming the costs of REIs to traditional energies.

## 4.2 Renewable energy efficiency

To see the prediction of renewable energy factors as per Eq. (1), we focused on the future development of China. As shown in Fig 2, solar growth (SG), wind growth (WG) and biomass and other generation (GBO) show their trends from 2012–2021. Among the '3' major renewable energy technologies (RETs), the rising rate of wind will be fastest followed by the SG and GBO. Fig 2, presents that the average growth of wind from 2012–2021 would be 28.67%, that of solar would be 34.88% and that of GBO would be 22.03%. For example, China became the nation with the world's largest installed capacity in 2011 and the biggest solar PV capacity in 2015, thus, in 2018, China added 35% and 33% of the worldwide overall installed capacity of wind and solar [76].

The share of individual renewable energy technology will change during the coming decade because China has taken serious initiatives to fight pollution. As shown in Fig 3, BAU, RS and AS scenarios, among the major RETs, the consumable capacity added by wind would increase (BAU) from 103.05 TWh in 2012 to about 109 TWh in 2021 in the AS scenario. However, in the SG scenario, the share of energy will grow double from 2012–2021 to more than 83%, 96% and 99%. With the fastest growth of solar energy during the next decade, its share would rise more than 4 times by 2021, arriving 98% under the RS scenario and 99% under the AS scenario. Moreover, as shown in Fig 3, there are some difficulties in attaining 2021 objectives under the RS scenario. Thus, the authorities must improve the proportion of RETs and add a suitable technological roadmap. However, to add the share of RETs in overall energy consumption, it is necessary to investigate its major factors, such as population, industry, technology, and environment. The numerator is the capacity to substitute traditional fuels with renewables [77].

## 4.3 Discussion

**4.3.1. Assessment of renewable energy enterprise.** As per the market fundamentalist and economic rationalist analysis of the influence of clean energy, the enterprise is the debate that growing clean energy resources using these programs is more expensive for end-users than another resource of getting new energy supplies. It can be noted that these initiatives give maximum amounts to renewable energy sellers than they would be capable of attaining either selling into a modest extensive electricity market. For instance, China adds most of the upward forecast revisions for 2022 and 2023, notwithstanding the phase-out of inducements for all

**Table 2. Electricity rates influence REIs in China.**

| Author | Method | Design and outcomes | Discourses |
|---|---|---|---|
| [28] | Politicized assessment of energy deployment method | Measured and explored the influence of media discourse on societal insights towards biomass energy consumption in various economic zones. They found that the effect of media discourse on social observations of biomass energy is exposed | Social perceptions / widened the impact |
| [69] | Multi-level perspective and multi-actor perspective theories are applied | The study applies the barriers to microgrid expansion applying a case study of a pilot zone in Qingdao. The results show that innovators in China's microgrid expansion and grasp insinuations for legislators to make more focused policy mixes to assist energy transition efforts. | Rationalist in development |
| [70] | Flexible collection information | Measures tests to China's transition to a less carbon electricity structure. In this system, the renewable energy would play an important role. Resilience of system can still be improved by the expansion of conventional ways and monitoring tackles and methods, for example, evaded cost basis for energy productivity investments, cohesive planning to develop the organization of generation, broadcast, and demand-side investments, and an obvious rate-making procedure. | Environmental modernist |
| [71] | Solar energy for poverty alleviation programme | An ambitious plan to support and ease rural poverty by organizing distributed solar photovoltaic (SEPAP) systems in poor areas of China (Beijing Qinghai province). Results show that the restraints on applying SEPAP and challenged local perspectives on the build-out of apparently low-carbon infrastructure for electricity generation. | Ecological and economic rationalist |
| [72] | Generalized Method of Moments | Using the provincial information involving the overall period of the Feed-In Tariff (FIT) support tool, this study discovers renewable energy subsidies' impact, prices and related economic influences on the fixed capacity of utility-scale solar PV in China. The results show that the FIT appliance in cycle with an important decline in REI costs has been the main driver of China's dramatic rise in utility-scale solar investments. | Significant relationship between energy, economy and cost |
| [73] | Causality analysis | This research takes solar PV in China as a model and uses a difference-in-difference structure that influences China's zonal FIT policy strategy and its manifold variations in future. The results show that the price of $CO_2$ emissions reduction by PV FITs is measured at about 120 yuan/ton of $CO_2$. | Environmental and economic rationalist |
| [74] | Accounting cost method | This study will re-evaluate China's solar resource endowments into five regions. The outcomes show that FIT is reasonable and real, and the government should regulate the FIT more regularly as per the FIT pricing model in this study. | Economic and cost rationalist |
| [75] | Net Present Value and Real Option methods | The study attains an equilibrium between dropping the financial load on the administration and guaranteeing the productivity of investors to rationalize the provincial differences in China. Results show that the FIT level is accessible as a price range based on a certain Internal Rate of Return that lies between 8%-15% for the investors of wind power. The present FIT price should be realigned and reallocated. | Cost and investment rationalist |

renewable energy in the previous year. This development is because of the four major different market and government factors [78]. First, the majority of the provinces' solar PV generation costs and onshore wind are lesser than coal standard costs. Second, the government declared 450 gigawatts of extra widespread solar PV and onshore wind megaprojects in Xinjiang and Inner Mongolia provinces, which are called "mega hubs". These projects are based on 100 gigawatts initiating growth at the start of 2022. Third, the Ministry of Finance declared the payment of USD 60 billion for renewable energy subsidies to be paid through 2022, enlightening the balance sheet of designers and providing extra money for coming projects. Finally, without national support, regional administrations are still giving tax inducements and less-price financing to renewable energy-related projects. Generally speaking, renewable energy sources are more costly in respect of total capital and operating costs than their fossil fuel competitors.

Consequently, REIs lead to energy prices that are greater than they might else be, for instance, few analysis emphasize the influence of these maximum prices on consumers [79–81]. While, some of the authors analyzed that advanced energy prices in turn have an adverse

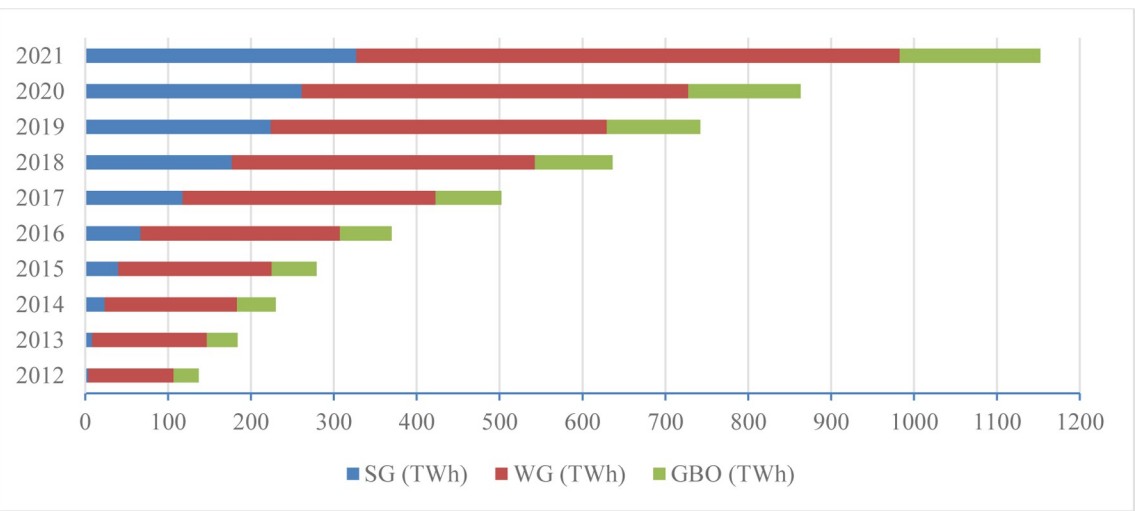

**Fig 2. Substitution to traditional energy under RS from 2010–2021.** *Source*: Data are published by China Statistical Yearbook and World Bank.

influence on the economy as a whole, decelerating the speed of economic progress, which is then interpreted into negative employment influences. This will offset any advancement in the renewable energy sector [82–85]. Moreover, energy communities play an imperative role in sustainable development, for instance, in 2015, the United Nations approved the proposal of seventeen sustainable development goals (SDGs) based on affordability, safety, sustainability and modern energy. The objective was to take action to challenge climate change and its effects. Similarly, a study by Losada-Puente et al. [86] investigated the multi-country analysis for Italy, Greece, and Spain based on the survey method. They estimated that the renewable energy legislation has not been finished in Southern Europe. Thus, the legal, economic,

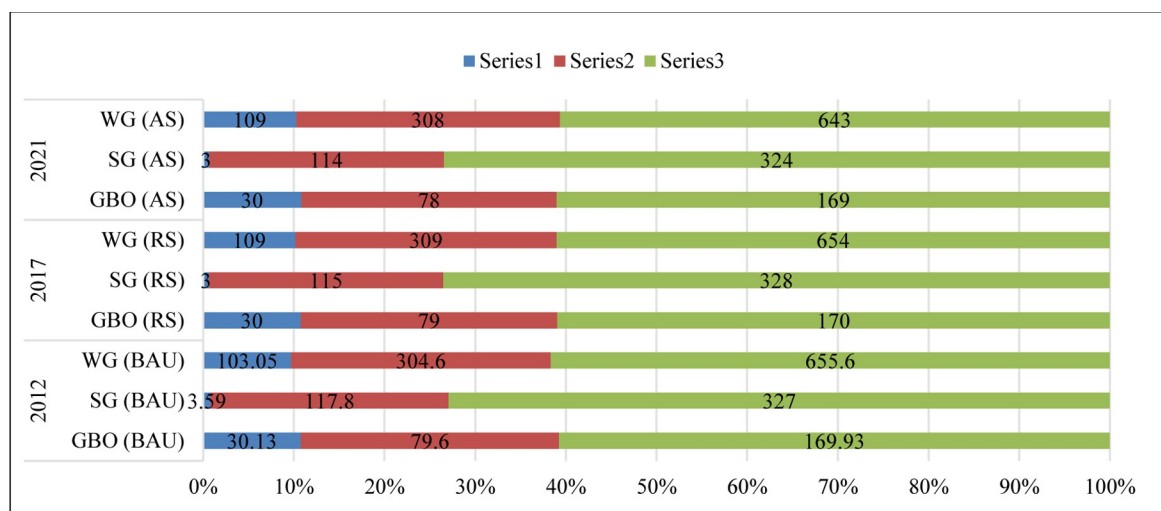

**Fig 3. Proportion of capacity to substitute traditional energy by renewable energy technology (Unit: TWh).** *Note*: series 1 denote the wind, solar and geothermal, biomass and other generation (blue color) during 2012, 2017 and 2021. Series 2 denote the wind, solar and geothermal, biomass and other generation (red color) during 2012, 2017 and 2021 and series 3 denotes the wind, solar and geothermal, biomass and other generation (green color) during 2012, 2017 and 2021. All the series are estimated at BAU, AS and RS scenarios.

administrative, technical and organizational perspectives of energy communities in these countries can be adopted, which are consistent with other countries [e.g., [34, 37, 43, 87]].

**4.3.2. Renewable energy promoters and modeling.** The replies to these investigations from clean energy supporters have also been comparatively reliable through the authorities studied and replicated more environmental development and advanced political economy, respectively. Moreover, regarding the matters over modeling framework and methods, researches emphasize the significance of ascribing value to the ecological advantages of clean energy sources, the requirement to be precise for persistent policy and market misrepresentations for fewer supportable technologies and economic growth efficiency of clean energy industries [88, 89].

From the perspectives of modeling issues, the computable general equilibrium (CGE), Environmental Kuznets Curve (EKC), and regression analysis are applied for the aims of modeling the effects of REIs, which may integrate rules that any interference in markets by the country will produce negative outcomes. The CGE modeling considers that markets are in balance and that costs are Pareto optimal but, in this case, the results in the market create distortions and lessen social welfare. No doubt, this method is appropriate for knowing how comparative rates vary when policy interference is made. Thus, in the case of renewable energy, the supposition of Pareto optimal values can result in letdowns where the loss is caused by carbon dioxide emissions or related ecological issues. Therefore, subsidies and externalities are necessary to make renewable energy policies to raise or decline community well-being, for example, the enhancement of the environment. In the previous models, for example, Pesaran et al. [90], Grossman and Krueger [91] and Böhringer et al. [92] analyzed the paucity of externalities from their models. The applications of this gap may not completely discuss or comprehended by policy-makers. Moreover, few technical discussions of how the models applied to measure the effects of REIs treat job creation in various divisions, for example, Raza and Lin [87] used a production model to estimate Pakistan's renewable energy technologies, Raza and Song [93] for Pakistan's energy substitution and environmental issues and Li and Ho [94] for 121 countries to check the environmental regulations using the heterogeneity analysis. They found how the models applied to measure the influence of REIs on the development of the export market for RETs in domestic demand.

The price assessment applied by FIT critics in China has been based on comparatively simple extensions of the specified areas' objectives for renewable energy at FIT rates in comparison to similar amounts of energy by traditional energy sources, i.e., coal, gas and oil. Moreover, using the dynamic modeling of the provincial electricity framework to evaluate how renewables would truly be incorporated and applied in the system, thus, methods could add to the perspective for solar PV to offset maximum price peaking supply from oil imports or gas-fired power plants. Solar PV during the daytime and wind during the overnight might also be able to counterbalance each other and lessen the demand for dispatchable backup [7, 95]. As per the literature, simple methods have been employed to measure energy cost impact of renewable energy in other jurisdictions in China and as a whole. Finally, the analysis found that the influence of the FIT scheme on renewable electricity prices would be marginal against the existing alternatives, mainly coal and natural gas in the case of China.

**4.3.3. Renewable energy costing, economy and subsidies.** Many of the scholars have discussed energy, economic and environmental issues few are particular to the situations of sole jurisdictions, while a maximum of them are of more general application. From China's perspective, renewable energy supporters have found that the market and electricity generation by the overall market. For instance, a common idea of evaluation with renewable costs is recognized by the FIT program [53]. As shown in Table 1, renewable hydropower is the highest in cost while solar PV is the second in cost, which shows that the government is spending a lot on

renewable electricity. In respect of FIT price per kilowatt hour shows the maximum rate on solar PV, wind onshore, and wind offshore, biogas, and solid biofuels, which are consistent in reducing pollution by 12 Mt, 5.6 Mt, 5.3 Mt, and 1.8 Mt during the current period. RETs in China have largely been led by FIT, set by the federal government and modified periodically. This tool led to extraordinary development, but needed enough flexibility to answer to cost variations and fixed subsidies for more cost reduction. To expedite cost reduction, an auction tool was initiated to license projects with lesser dependence on subsidies [77].

In reality, most of the new renewable energy plants, such as solar and wind since 1990 and biofuels since 2002 are based on price agreements well more than the market value. Therefore, it is essential to consider the capital prices of new buildings, and the need to give a satisfactory return on investment to attract private companies. As shown in Fig 4, the maximum share of the national adjustment of energy is related to coal by 63% and gas by 3%, while the renewable energy portion is 44% which is increasing. In order to compare the real economic value of efficient sources of new supply, capital, operating and running costs over the anticipated life of the project should be measured. Consequently, in favor of technological patents, solar PV is on the top figures (169,460), which has provided a significant impact in reducing carbon emissions by 12 Mt in 2021, which proves that renewable resources are higher sustainable for the economy and environment.

From the subsidies perspective, the impact of RET initiatives has been to query the action of externalized prices and risks linked with conventional energy supply, which are evaded in the sources of RETs. Actually, these costs might involve the life-cycle ecological and social influences of fossil fuels. Moreover, renewable energy supporters emphasize the effect of previous subsidies for the growth of traditional technologies, especially coal, oil and gas. Past studies related to renewable energy show that it is necessary to rise the proportion of renewable energy research and development and to improve the transformation of the energy mix [96], but subsidies are also compulsory because of the intrinsic shortcomings of the sector. Thus, internationalizing the optimistic externalities of clean energy is the key to subsidies.

In the end, the economic costs, such as capital and operating, excluding environmental and societal costs and risks, have been incredible to attain in the face of the traditional help for traditional technologies. For the long-term accessibility and infrastructure, particularly grid transmission and supply of technologies should be enhanced (i.e., RETs, FITs, and Jurisdiction involvement). However, all the frameworks should be politically viable, environmental modernists substitutes to discourse these fixed biases in the energy system framework. All the FITs should be advantageous to individuals and the public, financial and institutional size should be added to handle the transaction charges and bidding processes [97].

## 5. Conclusion and policy recommendations

### 5.1. Conclusion

The study objective was to present how renewable energy, economy, feed-in-tariff (FIT), cost, and environment have been designed within China's national energy policy discourse in the initial period to the date of its development. Followers of REIs like the China FIT scheme say that they offer the possibility to carry more ecologically sustainable, less costly energy supply security. However, the development of indigenous RETs and the application of renewable energy in various sectors are impressive. Based on the latest empirical information, including economic, employment, technology, cost, carbon emissions, and overall energy in China is extremely limited to analysis. Lacking consistent and comprehensive data about these factors in China has been grounded in the outcomes of modeling practices but no one has debated in discourse perspectives. Therefore, knowing the assumptions surrounding the frameworks

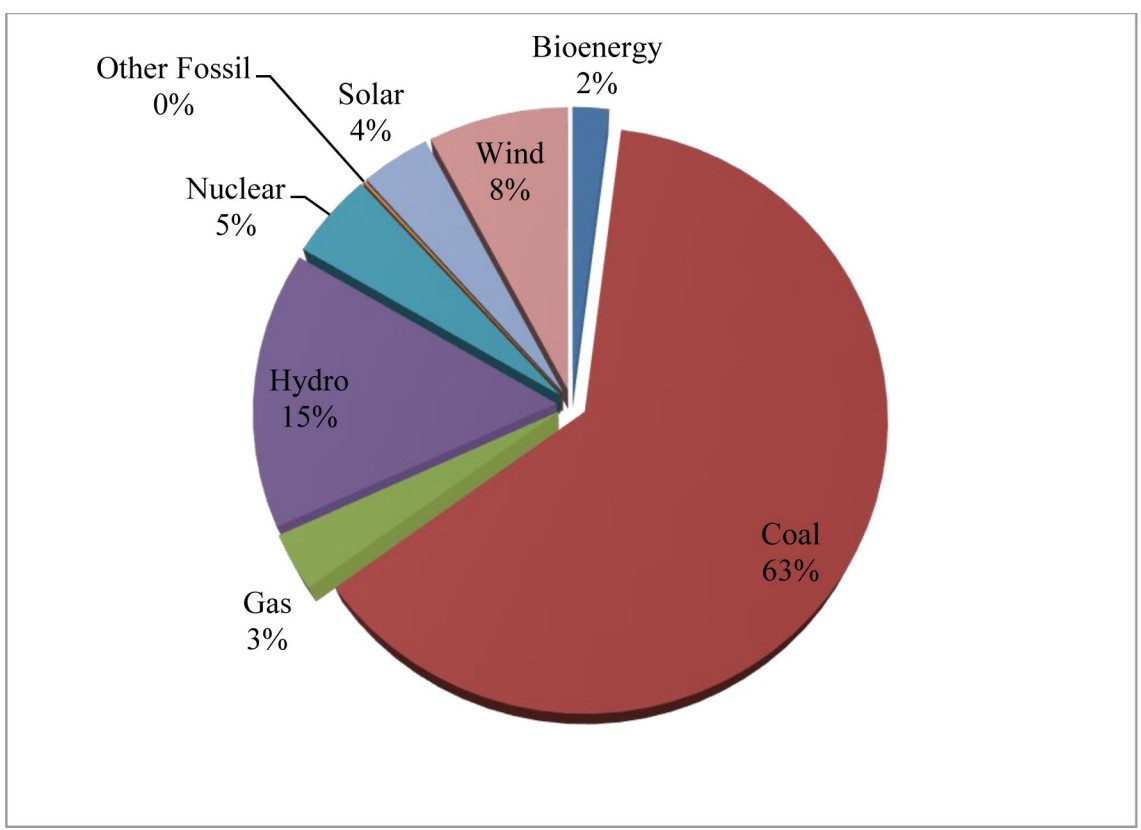

**Fig 4. China's fossil fuel and renewable electricity generation until August 2022.**

applied to analyze the influence of legislation is central to understanding the various outcomes influenced by contributors in the discussion over its impacts. Thus, some imperative conclusions can be found from the relative investigation of the discussions encompassing the economic growth impact of China's energy and green economy.

First, as per literature and other jurisdictions studied, advice about the effects of REIs on the economy as a whole is greatly hypothesized on assumptions. This shows that the renewable energy will cost higher than traditional and fossil fuel resources. The reason is that, RETs and generation are perceived as integrally more costly than the presented resources. As the government acknowledged 450 gigawatts of additional extensive solar PV and onshore wind megaprojects in Xinjiang and Inner Mongolia provinces, which are called "mega hubs". These projects are the better findings of RES in China.

Second, it is debated that REIs, for example, feed-in-tariff and renewable standards, conclude in maximum costs and energy prices to consumers rather than competitive processes for getting innovative energy supplies. The upper prices feed into the broader economy, lessen economic development and impact on country's employment. Thus, regarding FIT price/kilowatt hour expresses the determined proportion of solar PV, wind onshore, and wind offshore, biogas, and solid biofuels, which are reliable in falling contamination by 12, 5.6, 5.3, and 1.8 Mt. RETs in China have mainly been directed by FIT, set by the federal government and adapted occasionally.

Third, the supporters of REIs claim that FIT and similar schemes are a politically viable way of allocating subsidies, risks and externalities linked with fossil fuel sources. This is because the government has the power to overlook and design the market system. However, the cost is the

major factor, when these influencing factors are taken into consideration (i.e., environment, RETs, energy substitution and supply), which is also concerned with the suitable capital and operating costs of various technologies. Due to the quick growth of solar PV in the next decade, its proportion will grow higher than four times by 2021, reaching 98%. However, these RET impacts are rooted in broader discourses about the possible roles of governments and markets in progressing economic growth and environmental stability.

Fourth, various modeling and approaches have been applied that reflect various conceptual and theoretical ideas on the academic part. We found that market fundamentalists found negative and positive impacts under renewable energy and RETs, and have found the questions on cost, supply and avoiding the externalities. For this, the RE helpful framework was documented; giving for related environmental sustenance of RE plans, and lessening the subsidiary costs from the government. Hence, China will contribute 40% of worldwide cleaner volume growth between 2019–2024, run by improved system combination, inadequate cost and larger competitiveness of solar and wind. Overall, researchers found that ecological modernism is linked with RETs, providing the significance of the need to advance sustainability in the standard structure. Thus, following these perspectives can bring imperative insights, for example, sustainability between environmental modernism, and political economists' development plays an important role in future strategies.

Finally, based on the Grey Prediction Model and scenario analysis, the prediction and technological road map in '3' scenarios are presented for RETs. Under the aggressive scenario, solar and wind energy capacity can replace traditional energy. By 2021, the share of RETs in total energy production arrived at 643 TWh and 624 TWh, however, solar prediction is the fastest in growth. thus, the share of energy-conserving contribution among '3' major energy technologies is varying incessantly.

## 5.2. Policy recommendations

Since 1992, China has introduced a series of renewable energy policies, industrial planning and RETs for economic and environmental sustainability. The policy discourse system of China's renewable energy has been gradually formed and laid the sound foundation for China's energy and economic development. To fulfill the energy demand and enhance energy saving and $CO_2$ emissions mitigation, the medium and long-term development plan for renewable energy in 2007 explained the general aims for the RETs in the coming two decades by the implications of a fixed FIT, market quota, investment subsidies, and lessen the tax to increase the level of renewables, which is in line with the analysis of Song et al. [96]. However, as per the National Development and Reform Commission and Energy Bureau planned their ideas on enhancing institutional systems, and policy processes for green and low-carbon energy transition to found a national unified electricity market design on the supply side, enhance the electricity substitution tool, national electricity market system, trading system based on market supply.

For this, under the design and implementation, the foundation for the province's FIT program should be enhanced. The growth of best understandings and methods to attributing economic value to the full range, such as environment, land, capital, labor, costs, and relevant risks linked with energy facility building, operation, and fuel cycles are very imperative to carry-out the valuations of related factors and renewable energy sources relative to fossil fuels. Moreover, China's renewable energy policy will focus on reducing consumption and emissions and developing new energy sources under the Dual Carbon Goals (14th Five-Year Plan for Renewable Energy Development). Hence, the debate for China suggests several productive outputs in future exercises to support an understanding of the economic and ecological

influences of REIs. From the energy cost perspective, the development of RETs and energy sources using REIs could be applied as key factors in wider models of the economy. The study includes a few limitations: it was highlighted that apart from environmental, juridical limitations exist regarding individualities of provincial and ownership rights, consumption and RETs as per population growth. Regarding data limitations, most of the information on RETs and Five-Year development plans, the data can be analyzed on RET adoption at local levels, such as provincial and district.

## Supporting information

**S1 Data. Analysis sheet as supplementary data.**
(XLSX)

## Author Contributions

**Conceptualization:** Muhammad Yousaf Raza.

**Data curation:** Baohong Jiang, Muhammad Yousaf Raza.

**Formal analysis:** Baohong Jiang, Muhammad Yousaf Raza.

**Funding acquisition:** Muhammad Yousaf Raza.

**Investigation:** Muhammad Yousaf Raza.

**Methodology:** Baohong Jiang, Muhammad Yousaf Raza.

**Software:** Baohong Jiang, Muhammad Yousaf Raza.

**Supervision:** Baohong Jiang, Muhammad Yousaf Raza.

**Validation:** Baohong Jiang, Muhammad Yousaf Raza.

**Visualization:** Muhammad Yousaf Raza.

**Writing – original draft:** Baohong Jiang, Muhammad Yousaf Raza.

**Writing – review & editing:** Baohong Jiang, Muhammad Yousaf Raza.

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
