## [Decision Letter · Decision Letter 0]

26 Dec 2023

PONE-D-23-41388Renewable energy for sustainable development in China: Discourse analysisPLOS ONE

Dear Dr. RAZA,

Thank you for submitting your manuscript to PLOS ONE. After careful consideration, we feel that it has merit but does not fully meet PLOS ONE’s publication criteria as it currently stands. Therefore, we invite you to submit a revised version of the manuscript that addresses the points raised during the review process.

We look forward to receiving your revised manuscript.

Kind regards,

Grigorios L. Kyriakopoulos, 2 PhDs, 3 MSc, 2 MA, MEng, 2 BA, BSc

Academic Editor

PLOS ONE

Journal Requirements:

2. Please clearly specify the data sources in the Methods section.

   "This paper is supported by the China, the National Social Science Fund of China (Grant No.21BJY113)."

   "This paper is supported by the China, the National Social Science Fund of China (Grant No.21BJY113)."

   "This paper is supported by the China, the National Social Science Fund of China (Grant No.21BJY113)."

6. In the online submission form, you indicated that [Can be made available upon request]. 

7. PLOS requires an ORCID iD for the corresponding author in Editorial Manager on papers submitted after December 6th, 2016. Please ensure that you have an ORCID iD and that it is validated in Editorial Manager. To do this, go to ‘Update my Information’ (in the upper left-hand corner of the main menu), and click on the Fetch/Validate link next to the ORCID field. This will take you to the ORCID site and allow you to create a new iD or authenticate a pre-existing iD in Editorial Manager. Please see the following video for instructions on linking an ORCID iD to your Editorial Manager account: https://www.youtube.com/watch?v=_xcclfuvtxQ

Reviewers' comments:

Reviewer's Responses to Questions

**Comments to the Author**

1. Is the manuscript technically sound, and do the data support the conclusions?

Reviewer #1: Yes

Reviewer #2: Yes

Reviewer #3: Yes

2. Has the statistical analysis been performed appropriately and rigorously? 

Reviewer #1: Yes

Reviewer #2: Yes

Reviewer #3: Yes

3. Have the authors made all data underlying the findings in their manuscript fully available?

Reviewer #1: Yes

Reviewer #2: Yes

Reviewer #3: Yes

4. Is the manuscript presented in an intelligible fashion and written in standard English?

Reviewer #1: Yes

Reviewer #2: Yes

Reviewer #3: No

5. Review Comments to the Author

Reviewer #1: The present paper is highly interesting and is a major contribution to scientific progress. The paper deals with an interesting topic, especially for RES. This paper represents a good attempt.

This paper is well organised. It has a good structure and provides an easy and meaningful reading. The English writing is acceptable.

I recommend the publication of the paper under the following minor revisions:

1. It is necessary to be included further references related with the role OF RES

E.g.

Zerva, A., Tsantopoulos, G., Grigoroudis, E and Arabatzis, G. (2018). "Perceived citizens’ satisfaction with climate change stakeholders using a multicriteria decision analysis approach". Environmental Science and Policy, 82:60-70.

Kyriakopoulos, G.L., Arabatzis, G., Tsialis, P., Ioannou, K. (2018). "Electricity consumption and RES plants in Greece: Typologies of regional units". Renewable Energy. 127, pp. 134-144.

Eftaxias, A., Passa, E.A., Michailidis, C., Daoutis, C., Kantartzis, A., Diamantis, V. Residual Forest Biomass in Pinus Stands: Accumulation and Biogas Production Potential. Energies, 2022, 15(14), 5233.

Arabatzis, G and Malesios, Ch. (2013). "Pro-Environmental attitudes of users and not users of fuelwood in a rural area of Greece”. Renewable and Sustainable Energy Reviews, 22: 621-630.

Tampakis, S., Arabatzis, G., Tsantopoulos, G and Rerras, I. (2017). "Citizens' views on electricity use, savings and production from renewable energy sources: A case study from a Greek Island". Renewable and Sustainable Energy Reviews, 79: 39-49.

Zografidou, E., Petridis, K., Petridis, N and Arabatzis, G. (2017). "A financial approach to renewable energy production in Greece using goal programming". Renewable Energy, 108: 37–51.

Reviewer #2: I have had the opportunity to review your manuscript titled "Renewable Energy for Sustainable Development in China: Discourse Analysis," and I commend you on the thorough exploration of the subject matter. Overall, I find the study to be well-structured, with clear and concise language, demonstrating a commendable level of proficiency in English. The manuscript holds scientific value and provides valuable insights into the economic and environmental impacts of renewable energy initiatives in China.

Your comprehensive literature review demonstrates a solid foundation for the study, but I would suggest enhancing it further, particularly in relation to the broader context of the renewable energy strategy. Expanding on this aspect could strengthen the theoretical framework and provide readers with a more comprehensive understanding of the research landscape.

I appreciate the rigorous research methods employed in the analysis of data, which contributes to the credibility of your findings. The paper effectively addresses the discourse surrounding renewable energy for sustainable development in China, making it relevant for policymakers and researchers in the field.

Moreover, I recommend enriching the conclusions section with additional findings to provide a more robust summary of the key takeaways from your study. This could enhance the overall impact of your research and make it more valuable for readers seeking insights into the subject matter.

In conclusion, I believe that with these minor revisions, your manuscript can achieve an even higher level of quality. I appreciate the effort you have put into your work and hope that my comments prove helpful in refining your manuscript. I recommend the publication of your study, anticipating that it will contribute meaningfully to the discourse on renewable energy and sustainable development in China.

Reviewer #3: Your manuscript offers an insightful exploration into the RES development in China. For future revisions to further enhance the quality and significance of your research, I suggest considering the following improvements:

1. The language in some parts of the text requires enhancement. For example, in page 4, the sentence “In addition, the focus has been on national climate change and mitigation of carbon emissions by applying renewables…”. What is “national climate change”?

2. So, please make a thorough review of the manuscript for language improvement.

3. According to https://chineseclimatepolicy.oxfordenergy.org/book-content/domestic-policies/renewable-power/ feed-in tariffs are being replaced with a range of policies and market mechanisms. The afore mentioned website contains a number of such policies and mechanisms. Please reflect this information in the introduction and in your analysis.

4. Considering the pressing issue of our times is climate change, and that any reduction of the clinker content by e.g. ash from banana leaves has the aim to reduce energy consumption and GHG emissions, please consider linking the introduction of the paper to global warming and climate change. This approach not only contextualizes your topic within a global challenge but also offers a comprehensive perspective on the role of the replacement of clinker with ash from banana leaves in climate mitigation action plans. Reference these studies for support:

• Akkermans et al (2023). Exploring long-term mitigation pathways for a net zero Tajikistan. Mitigation and Adaptation Strategies for Global Change, 28(3), 1-26. https://doi.org/10.1007/s11027-023-10053-w

• Sebos et al (2020). Methodological framework for the quantification of GHG emission reductions from climate change mitigation actions. Strategic planning for energy and the environment, 219-242. doi: 10.13052/spee1048-4236.391411

• Kyriakopoulos et al (2023). Enhancing Climate Neutrality and Resilience through Coordinated Climate Action: Review of the Synergies between Mitigation and Adaptation Actions. Climate, 11(5), 105. https://doi.org/10.3390/cli11050105

• Kyriakopoulos et al (2023). Benefits and Synergies in Addressing Climate Change via the Implementation of the Common Agricultural Policy in Greece. Applied Sciences, 13(4), 2216. https://doi.org/10.3390/app13042216

5. Which are the main stakeholders in decisions about developing RES in China? I consider that a discussion about the mapping and analysis of stakeholders influencing relevant actions and policies, along with their perceptions, would add value to the paper. This approach can provide a comprehensive view of the various factors at play. Refer to:

• Sebos et al. Stakeholder perceptions on climate change impacts and adaptation actions in Greece. Euro-Mediterr J Environ Integr (2023). https://doi.org/10.1007/s41207-023-00396-w

• Ioanna et al. Stakeholder mapping and analysis for climate change adaptation in Greece. Euro-Mediterr J Environ Integr 7, 339–346 (2022). https://doi.org/10.1007/s41207-022-00317-3

6. For a broader perspective, consider examining co-design principles in formulating carbon mitigation policies, which may offer insights into collaborative approaches for climate neutrality as well as resilience. Please refer to the study:

• Sebos et al: Climate-Resilient Urban Regeneration: Transforming School Yards for a Sustainable and Adaptive Future, EMS Annual Meeting 2023, Bratislava, Slovakia, 4–8 Sep 2023, EMS2023-73, https://doi.org/10.5194/ems2023-73, 2023

7. Please enrich the “Discussion” by adding innovative mitigation options, such as energy communities. You may consult the paper:

• Losada-Puente et al. Cross-Case Analysis of the Energy Communities in Spain, Italy, and Greece: Progress, Barriers, and the Road Ahead. Sustainability 2023, 15, 14016. https://doi.org/10.3390/su151814016

8. It would be beneficial for the paper to enhance the conclusion by providing a more detailed discussion of your study's limitations and uncertainties. This transparency will add to the credibility and scope for future research directions.

6. PLOS authors have the option to publish the peer review history of their article (what does this mean?). If published, this will include your full peer review and any attached files.

Reviewer #1: No

Reviewer #2: No

Reviewer #3: **Yes: **Ioannis Sebos

---

## [Author Response · Author response to Decision Letter 0]

17 Jan 2024

Response to Reviewers’ Comments

The authors are thankful to editor and reviewers for their valuable comment on the previous version of their manuscript. Based on these comments, the authors have thoroughly revised the entire manuscript in accordance with comments from anonymous reviewers. Indeed, these comments have proven very useful for the improvement of our paper. Our point-by-point responses, as documented for each reviewer, are shown below (For ease of reading, the reviewer’s original comments are provided in Blue). For easy identification of the changes we made, we present both a marked-up version (blue color) and a clean version of the revised manuscript. We have answered to both to reviewer’s questions and comments on manuscript. Also, red highlights in the marked version of manuscript are the language corrections.

EDITORIAL

1. Please upload an updated copy of your revised manuscript that does not contain any tracked changes or highlighting as your main article file. This will be used in the production process if your manuscript is accepted. Please amend the file type for the file showing your changes to Revised Manuscript w/tracked changes. Please follow this link for more information: http://blogs.PLOS.org/everyone/2011/05/10/how-to-submit-your-revise-manuscript/

Response: Added both marked and unmarked revised copies.

Please remove any unnecessary or old files from your revision, and make sure that only those relevant to the current version of the manuscript are included.

2. Please clearly specify the data sources in the Methods section.

Response: Added as well in line 253-254.

3. Please note that funding information should not appear in the Acknowledgments section/Funding section or Any other areas of your Manuscript. We will only publish funding information present in the Funding Statement section of the online submission form. Please remove any funding-related text from the manuscript.

Response: Deleted.

Response: Deleted

 "This paper is supported by the China, the National Social Science Fund of China (Grant No.21BJY113)."

Response: Deleted.

6. In the online submission form, you indicated that data "Can be made available upon request".

Response: Provided as well.

7. PLOS requires an ORCID iD for the corresponding author in Editorial Manager on papers submitted after December 6th, 2016. Please ensure that you have an ORCID iD and that it is validated in Editorial Manager. To do this, go to ‘Update my Information’ (in the upper left-hand corner of the main menu), and click on the Fetch/Validate link next to the ORCID field. This will take you to the ORCID site and allow you to create a new iD or authenticate a pre-existing iD in Editorial Manager. Please see the following video for instructions on linking an ORCID iD to your Editorial Manager account: https://www.youtube.com/watch?v=_xcclfuvtxQ

Response: Updated as well.

We've returned your manuscript to your account. Please resolve these issues and resubmit your manuscript within 21 days. If you need more time, please email the journal office at plosone@plos.org. We are happy to grant extensions of up to one month past this due date. If we do not hear from you within 21 days, we will withdraw your manuscript.

Reviewer's Responses to Questions

1. Is the manuscript technically sound, and do the data support the conclusions?

Reviewer #1: Yes

Reviewer #2: Yes

Reviewer #3: Yes

Response: Thank you very much.

2. Has the statistical analysis been performed appropriately and rigorously?

Reviewer #1: Yes

Reviewer #2: Yes

Reviewer #3: Yes

Response: Thank you very much.

3. Have the authors made all data underlying the findings in their manuscript fully available?

Reviewer #1: Yes

Reviewer #2: Yes

Reviewer #3: Yes

Response: Thank you very much.

4. Is the manuscript presented in an intelligible fashion and written in standard English?

Reviewer #1: Yes

Response: Thank you very much.

Reviewer #2: Yes

Response: Thank you very much.

Reviewer #3: No

Response: Authors are thankful to the reviewer. We have thoroughly edited the manuscript and also received help through a native colleague. Moreover, we have highlighted all the corrections in red color.

Responses to reviewer’s #1 comments

The present paper is highly interesting and is a major contribution to scientific progress. The paper deals with an interesting topic, especially for RES. This paper represents a good attempt.

This paper is well organised. It has a good structure and provides an easy and meaningful reading. The English writing is acceptable.

I recommend the publication of the paper under the following minor revisions:

Response: Authors are thankful to the reviewer for appreciating our study.

1. It is necessary to be included further references related with the role OF RES

E.g.

Zerva, A., Tsantopoulos, G., Grigoroudis, E and Arabatzis, G. (2018). "Perceived citizens’ satisfaction with climate change stakeholders using a multicriteria decision analysis approach". Environmental Science and Policy, 82:60-70.

Kyriakopoulos, G.L., Arabatzis, G., Tsialis, P., Ioannou, K. (2018). "Electricity consumption and RES plants in Greece: Typologies of regional units". Renewable Energy. 127, pp. 134-144.

Eftaxias, A., Passa, E.A., Michailidis, C., Daoutis, C., Kantartzis, A., Diamantis, V. Residual Forest Biomass in Pinus Stands: Accumulation and Biogas Production Potential. Energies, 2022, 15(14), 5233.

Arabatzis, G and Malesios, Ch. (2013). "Pro-Environmental attitudes of users and not users of fuelwood in a rural area of Greece”. Renewable and Sustainable Energy Reviews, 22: 621-630.

Tampakis, S., Arabatzis, G., Tsantopoulos, G and Rerras, I. (2017). "Citizens' views on electricity use, savings and production from renewable energy sources: A case study from a Greek Island". Renewable and Sustainable Energy Reviews, 79: 39-49.

Zografidou, E., Petridis, K., Petridis, N and Arabatzis, G. (2017). "A financial approach to renewable energy production in Greece using goal programming". Renewable Energy, 108: 37–51.

Response: Authors are thankful to the reviewer for adding relevant information. On the basis, we have added these studies . please see line#104-114.

Responses to reviewer’s #2 comments

1. I have had the opportunity to review your manuscript titled "Renewable Energy for Sustainable Development in China: Discourse Analysis," and I commend you on the thorough exploration of the subject matter. Overall, I find the study to be well-structured, with clear and concise language, demonstrating a commendable level of proficiency in English. The manuscript holds scientific value and provides valuable insights into the economic and environmental impacts of renewable energy initiatives in China.

Response: Thank you very much for your appreciation.

2. Your comprehensive literature review demonstrates a solid foundation for the study, but I would suggest enhancing it further, particularly in relation to the broader context of the renewable energy strategy. Expanding on this aspect could strengthen the theoretical framework and provide readers with a more comprehensive understanding of the research landscape.

Response: Thank you very much. We have added the literature which is really supportive to our study. See line#104-114.

3. I appreciate the rigorous research methods employed in the analysis of data, which contributes to the credibility of your findings. The paper effectively addresses the discourse surrounding renewable energy for sustainable development in China, making it relevant for policymakers and researchers in the field.

Response: Thank you very much. Authors are thankful to the reviewer for appreciationg our study.

4. Moreover, I recommend enriching the conclusions section with additional findings to provide a more robust summary of the key takeaways from your study. This could enhance the overall impact of your research and make it more valuable for readers seeking insights into the subject matter.

Response: Thank you very much. We have added and strengthen the conclusion section based on empirical findings and assessment of Grey prediction scenarios. See line#505-508; 512-515; 521-522; 528-531

5. In conclusion, I believe that with these minor revisions, your manuscript can achieve an even higher level of quality. I appreciate the effort you have put into your work and hope that my comments prove helpful in refining your manuscript. I recommend the publication of your study, anticipating that it will contribute meaningfully to the discourse on renewable energy and sustainable development in China.

Response: Authors are grateful to the reviewers. We have tried our best to revise the recommendations properly. 

Responses to reviewer’s #3 comments

Your manuscript offers an insightful exploration into the RES development in China. For future revisions to further enhance the quality and significance of your research, I suggest considering the following improvements:

1. The language in some parts of the text requires enhancement. For example, in page 4, the sentence “In addition, the focus has been on national climate change and mitigation of carbon emissions by applying renewables…”. What is “national climate change”?

Response: Thank you very much. we have revised the sentence following your suggestion. See line#126-130.

2. So, please make a thorough review of the manuscript for language improvement.

Response: Thank you so much. We have thoroughly improved and edited the manuscript and highlighted in red color.

3.According to https://chineseclimatepolicy.oxfordenergy.org/book-content/domestic-policies/renewable-power/ feed-in tariffs are being replaced with a range of policies and market mechanisms. The afore mentioned website contains a number of such policies and mechanisms. Please reflect this information in the introduction and in your analysis.

Response: Authors are thankful to the reviewers for suggesting a valuable information regarding China. We have added the useful informations as well in both introduction and discussion section. See line#43-47; 2541-256.

4. Considering the pressing issue of our times is climate change, and that any reduction of the clinker content by e.g. ash from banana leaves has the aim to reduce energy consumption and GHG emissions, please consider linking the introduction of the paper to global warming and climate change. This approach not only contextualizes your topic within a global challenge but also offers a comprehensive perspective on the role of the replacement of clinker with ash from banana leaves in climate mitigation action plans. Reference these studies for support:

• Akkermans et al (2023). Exploring long-term mitigation pathways for a net zero Tajikistan. Mitigation and Adaptation Strategies for Global Change, 28(3), 1-26. https://doi.org/10.1007/s11027-023-10053-w

• Sebos et al (2020). Methodological framework for the quantification of GHG emission reductions from climate change mitigation actions. Strategic planning for energy and the environment, 219-242. doi: 10.13052/spee1048-4236.391411

• Kyriakopoulos et al (2023). Enhancing Climate Neutrality and Resilience through Coordinated Climate Action: Review of the Synergies between Mitigation and Adaptation Actions. Climate, 11(5), 105. https://doi.org/10.3390/cli11050105

• Kyriakopoulos et al (2023). Benefits and Synergies in Addressing Climate Change via the Implementation of the Common Agricultural Policy in Greece. Applied Sciences, 13(4), 2216. https://doi.org/10.3390/app13042216

Response: Thank you so much. We are thankful to you for providing useful information. On this basis, we hav added these studies. See line#130-132.

5. Which are the main stakeholders in decisions about developing RES in China? I consider that a discussion about the mapping and analysis of stakeholders influencing relevant actions and policies, along with their perceptions, would add value to the paper. This approach can provide a comprehensive view of the various factors at play. Refer to:

• Sebos et al. Stakeholder perceptions on climate change impacts and adaptation actions in Greece. Euro-Mediterr J Environ Integr (2023). https://doi.org/10.1007/s41207-023-00396-w

• Ioanna et al. Stakeholder mapping and analysis for climate change adaptation in Greece. Euro-Mediterr J Environ Integr 7, 339–346 (2022). https://doi.org/10.1007/s41207-022-00317-3

Response: Thank you very much. we have discussed the findings as well. See line#323-329.

6. For a broader perspective, consider examining co-design principles in formulating carbon mitigation policies, which may offer insights into collaborative approaches for climate neutrality as well as resilience. Please refer to the study:

• Sebos et al: Climate-Resilient Urban Regeneration: Transforming School Yards for a Sustainable and Adaptive Future, EMS Annual Meeting 2023, Bratislava, Slovakia, 4–8 Sep 2023, EMS2023-73, https://doi.org/10.5194/ems2023-73, 2023

Response: Thank you so much. We have further improved the significance of our study and added the study. Please see line#326-329.

7. Please enrich the “Discussion” by adding innovative mitigation options, such as energy communities. You may consult the paper:

• Losada-Puente et al. Cross-Case Analysis of the Energy Communities in Spain, Italy, and Greece: Progress, Barriers, and the Road Ahead. Sustainability 2023, 15, 14016. https://doi.org/10.3390/su151814016

Response: Thank you very much. we have enhanced the discussion section and made it consistent with relevant studies provided by you and discussed as well. See line#396-404.

8. It would be beneficial for the paper to enhance the conclusion by providing a more detailed discussion of your study's limitations and uncertainties. This transparency will add to the credibility and scope for future research directions.

Response: Thank you very much. We appreciate your concern. We have strengthened and added the future limitations of our study. See line#505-508; 512-515; 521-522; 528-531; 565-569.

Thank you

---

## [Decision Letter · Decision Letter 1]

23 Jan 2024

Renewable energy for sustainable development in China: Discourse analysis

PONE-D-23-41388R1

Dear Dr. Raza,

We’re pleased to inform you that your manuscript has been judged scientifically suitable for publication and will be formally accepted for publication once it meets all outstanding technical requirements.

Kind regards,

Grigorios L. Kyriakopoulos, 2 PhDs, 3 MSc, 2 MA, MEng, 2 BA, BSc

Academic Editor

PLOS ONE

Additional Editor Comments (optional):

Reviewers' comments:

Reviewer's Responses to Questions

**Comments to the Author**

1. If the authors have adequately addressed your comments raised in a previous round of review and you feel that this manuscript is now acceptable for publication, you may indicate that here to bypass the “Comments to the Author” section, enter your conflict of interest statement in the “Confidential to Editor” section, and submit your "Accept" recommendation.

Reviewer #1: All comments have been addressed

Reviewer #3: All comments have been addressed

2. Is the manuscript technically sound, and do the data support the conclusions?

Reviewer #1: Yes

Reviewer #3: Yes

3. Has the statistical analysis been performed appropriately and rigorously? 

Reviewer #1: Yes

Reviewer #3: Yes

4. Have the authors made all data underlying the findings in their manuscript fully available?

Reviewer #1: Yes

Reviewer #3: Yes

5. Is the manuscript presented in an intelligible fashion and written in standard English?

Reviewer #1: Yes

Reviewer #3: Yes

6. Review Comments to the Author

Reviewer #1: The present paper is highly interesting and is a major contribution to scientific progress. This paper is well organised. It has a good structure and provides an easy and meaningful reading.

All comments have been addressed.

Accepted.

Reviewer #3: All comments were addressed. Language of the paper has been improved.

7. PLOS authors have the option to publish the peer review history of their article (what does this mean?). If published, this will include your full peer review and any attached files.

Reviewer #1: No

Reviewer #3: **Yes: **Ioannis Sebos

---

## [Editor Report · Acceptance letter]

29 Feb 2024

PONE-D-23-41388R1 

PLOS ONE

Dear Dr. Raza, 

I'm pleased to inform you that your manuscript has been deemed suitable for publication in PLOS ONE. Congratulations! Your manuscript is now being handed over to our production team.

Kind regards, 

on behalf of

Dr. Grigorios L. Kyriakopoulos 

Academic Editor

PLOS ONE